# Magnetic Resonance Imaging Measurement of Entorhinal Cortex in the Diagnosis and Differential Diagnosis of Mild Cognitive Impairment and Alzheimer’s Disease

**DOI:** 10.3390/brainsci11091129

**Published:** 2021-08-26

**Authors:** Qianqian Li, Junkai Wang, Jianghong Liu, Yumeng Wang, Kuncheng Li

**Affiliations:** 1Department of Radiology and Nuclear Medicine, Xuanwu Hospital, Capital Medical University, Beijing 100053, China; qianqianli27@126.com; 2Being Key Laboratory of Magnetic Resonance Imaging and Brain Informatics, Beijing 100053, China; 3Department of Psychology, Tsinghua University, Haidian District, Beijing 100084, China; plzgg@mail.tsinghua.edu.cn; 4Department of Neurology, Xuanwu Hospital, Capital Medical University, Beijing 100053, China; 13341070223@189.cn; 5School of Psychology, Capital Normal University, Beijing 100048, China; 2193502061@cnu.edu.cn

**Keywords:** Alzheimer’s disease, amnestic mild cognitive impairment, entorhinal cortex measurements

## Abstract

Several magnetic resonance imaging studies have shown that the entorhinal cortex (ERC) is the first brain area related to pathologic changes in Alzheimer’s disease (AD), even before atrophy of the hippocampus (HP). However, change in ERC morphology (thickness, surface area and volume) in the progression from aMCI to AD, especially in the subtypes of aMCI (single-domain and multiple-domain: aMCI-s and aMCI-m), however, is still unclear. ERC thickness, surface area and volume were measured in 29 people with aMCI-s, 22 people with aMCI-m, 18 patients with AD and 26 age-/sex-matched healthy controls. Group comparisons of the ERC geometry measurements (including thickness, volume and surface area) were performed using analyses of covariance (ANCOVA). Furthermore, receiver operator characteristic (ROC) analyses and the area under the curve (AUC) were employed to investigate classification ability (HC, aMCI-s, aMCI-m and AD from each other). There was a significant decreasing tendency in ERC thickness from HC to aMCI-s to aMCI-m to finally AD in both the left and the right hemispheres (left hemisphere: HC > aMCI-s > AD; right hemisphere: aMCI-s > aMCI-m > AD). For ERC volume, both the AD group and the aMCI-m group showed significantly decreased volume on both sides compared with the HC group. In addition, the AD group also had significantly decreased volume on both sides compared with the aMCI-s group. As for the ERC surface area, no significant difference was identified among the four groups. Furthermore, the AUC results demonstrate that combined ERC parameters (thickness and volume) can better discriminate the four groups from each other than ERC thickness alone. Finally, and most importantly, relative to HP volume, the capacity of combined ERC parameters was better at discriminating between HC and aMCI-s, as well as aMCI-m and AD. ERC atrophy, particularly the combination of ERC thickness and volume, might be regarded as a promising candidate biomarker in the diagnosis and differential diagnosis of aMCI and AD.

## 1. Introduction

Alzheimer’s disease (AD) is one of the most common progressive neurodegenerative diseases. So far, there is no cure for AD. Early diagnosis, timely intervention and treatment can improve the prognosis of patients [1], and thus it is important to make a diagnosis in the preclinical stage.

Amnestic mild cognitive impairment (aMCI), a probable transitional stage between normal aging and early dementia, is associated with a high risk of developing AD [2]. However, the group of aMCI is a heterogeneous clinical entity [3]. Some MCI individuals convert to AD rapidly, some keep a stable state for many years, and others return to normal cognition [4]. Based on the patterns of cognitive impairment, aMCI can be classified into two subtypes: single-domain aMCI (aMCI-s), those with isolated memory impairment, and multiple-domain aMCI (aMCI-m), those with multiple cognitive domain decline, such as language, attention, visuospatial or executive function [5]. It has been suggested that subjects with aMCI-m are more likely to progress to AD than subjects with aMCI-s [3,6]. From this perspective, we reckon that aMCI-s may represent an earlier stage of aMCI and classification of aMCI subtypes is important to identify individuals at high risk for developing AD.

Morphological indexes, based on high-resolution structural magnetic resonance imaging (MRI), have become a standard method for the detection of incipient AD. It is well documented that volume loss of the hippocampus (HP) plays an important role in the early diagnosis of AD [7,8]. According to the new diagnostic criteria for MCI due to AD, MRI-based measures of medial temporal lobe (MTL) atrophy, especially HP, have been considered as important research criteria to enhance the specificity of the diagnosis [9]. However, several MRI studies have shown that the ERC is the first brain area related to pathologic changes of AD, even before HP atrophy occurs [10,11]. Moreover, the morphological indexes of the ERC generally included thickness, surface area and volume. These indexes have been derived on a special neurophysiological basis [12,13,14]. Generally, the thickness of the cerebral cortex likely represents neuron numbers and many other elements of the neuropil, such as dendrites, axons and so on [13,15]. Surface area, which is correlated with head size, may reflect local subcortical size [13]. As for the volume of the ERC, it is a composite measure associated with both thickness and surface area. To date, it is still unclear how ERC thickness, volume and surface area change among AD, aMCI and HC entities, especially for aMCI subtypes (aMCI-s and aMCI-m). In addition, different morphological indexes have been proven to be affected differently in the course of disease [13,16], and cortical thickness and volume were demonstrated to have differences among AD, aMCI and healthy controls in distinct cortical regions [17]. Therefore, combining multiple morphological indexes may effectively reveal subtle structural alterations in the early stage of AD and improve classification accuracy of aMCI subtypes.

In the current study, changes in ERC thickness, volume and surface area were first assessed among HC, aMCI-s, aMCI-m and AD participant groups. Then, the capacity of single ERC indexes (thickness, volume or surface area) or combined to discriminate the four groups from each other was separately identified. Based on previous studies, we hypothesized that combined ERC indexes may have an advantage over HP volume in detecting AD in the preclinical stage.

## 2. Materials and Methods

### 2.1. Participants

Sixty-nine participants were recruited from Memory Disorders Clinic and Department of Neurology, Xuanwu Hospital, Capital Medical University, including 29 people with aMCI-s (14 females, mean age 71.21 ± 6.4 years), 22 people with aMCI-m (10 females, mean age 71.09 ± 8.4 years) and 18 patients with AD (11 females, mean age 70.94 ± 9.7 years). Twenty-six age- and sex-matched healthy controls (15 females, mean age 70.38 ± 5.36 years) were recruited from community near Xuanwu Hospital. All participants or their guardians signed informed consent before participating in the study in accordance with the Institutional Review Board of Xuanwu Hospital, Capital Medical University, and gained compensation for their participation.

The clinical diagnosis of AD was established according to the Diagnostic and Statistical Manual of Mental Disorders-V (DSM-V) criteria for Alzheimer’s Dementia and the National Institute on Aging and Alzheimer’s Association (NIA-AA) Diagnostic Guidelines for AD [9,18]. aMCI participants were diagnosed and classified according to Petersen’s clinical diagnostic criteria and the NIA-AA criteria for MCI due to AD [5,9]. The exclusionary criteria included depression, seizures, stroke, Parkinson’s disease, neurosurgery history, head trauma and contraindications to MRI (e.g., aneurysm clip(s), cardiac pacemaker, any metallic fragment or foreign body, implanted cardioverter-defibrillator). Patient characteristics are summarized in Table 1.

All participants underwent a standard battery of neuropsychological tests that included the Clinical Dementia Rating (CDR) [19], the Mini-mental State Examination (MMSE) [20] and the Montreal cognitive assessment (MoCA) [21]. Four specific cognitive domains were assessed: (1) the visuospatial skill was measured with the clock-drawing test (CDT, 3-point) [22]; (2) the executive function was evaluated with the trail-making test (TMT) [23]; (3) language skill was assessed with the Boston naming test (BNT) [24]; (4) the memory function was evaluated with the Auditory Verbal Learning Test (Chinese version of AVLT) [25].

### 2.2. MRI Acquisition

Three-dimensional high-resolution structural T_1_-weighted images were obtained by magnetization-prepared rapid-acquisition gradient echo (MP-RAGE) pulse sequence on a 3.0 Tesla scanner (Trio Tim, Siemens, Medical Solutions, Erlangen, Germany). MP-RAGE indexes were as follows: repetition time (TR)/echo time (TE)/inversion time (TI)/flip angle (FA) = 1900 ms/2.2 ms/900 ms/9°, acquisition matrix = 224 × 256 × 176, voxel size = 1 × 1 × 1 mm^3^. The acquisition time was about 3 mins. Suitable foam padding was used to limit head movement, and earplugs were employed to minimize scanning noise.

### 2.3. MR Morphometric Image Analysis

The data were exported from scanner to a personal computer to perform morphometric analysis. Automatic segmentation of ERC was performed with FreeSurfer version 5.0, which is documented and freely available for download online (https://surfer.nmr.mgh.harvard.edu/, accessed on 20 March 2019). The technical details of FreeSurfer procedures were described elsewhere. Briefly, the processing included motion correction, removal of non-brain tissue, automated Talairach transformation, intensity normalization and estimation of gray matter/white matter boundary and pial surface. Cortical thickness was then defined as distance from the GM/WM boundary and the pial surface. The volume measures of left and right ERC were derived from the standard stats directory using the Desikan–Killiany atlas.

### 2.4. Statistical Analyses

SPSS (version 20.0, IBM, accessed on 10 May 2019) was utilized for statistical analyses. Demographic features were compared using one-way analysis of variance (ANOVA). Gender difference was tested using the chi-square test. Group comparisons of the left and right ERC were performed using analysis of covariance (ANCOVA), in which thickness, volume and surface area of the ERC were dependent variables, respectively. Gender, age and years of education were used as covariates. Moreover, head size, as estimated by estimated total intracranial volume (eTIV), was also considered as a covariate in all analyses for correcting head size variation in regional brain volume measurements. Although head size is not associated with thickness, the same covariates were used to attain formal statistical equivalency. *p* values of less than 0.05 were considered to indicate statistical significance. In addition, relationships between ERC morphometric measurements and cognitive domain scores were examined using partial correlation analyses, with age, gender and years of education as covariates with Bonferroni correction.

Furthermore, to investigate the classification ability (HC, aMCI-s, aMCI-m and AD from each other) of the ERC single index (including thickness, volume and surface area), ERC combined indexes (combining thickness, volume or surface area) or HP measure (volume), receiver operating characteristic (ROC) analyses, which show the diagnostic ability of a binary classifier as a function of its discrimination threshold, were performed. The area under the curve (AUC) was calculated and used as a differentiating indicator. MedCalc software (version 19, http://www.medcalc.org, accessed on 25 June 2019) was used for ROC curve analysis.

## 3. Results

Demographic features of patients and healthy controls are shown in Table 1. The four groups were comparable in terms of age (F (3, 91) = 0.064, *p* = 0.979) and gender distribution (χ^2^ = 1.461, *p* = 0.691). Education level (F (3, 91) = 9.342, *p* < 0.001) was significantly lower in the AD (*p* < 0.001) and aMCI-s (*p* = 0.001) groups than in the HC group. There were significant differences across the four groups in all cognitive measures (*p* < 0.001 for all). Specifically, based on post hoc comparisons, the AD group had the worst performance on all behavioral measures relative to the other three groups. The aMCI-m group showed significant impairment in all cognitive domains compared with the HC group and worse performances on BNT, TMT and CDT compared with the aMCI-s group. Moreover, the aMCI-s group showed significantly impaired cognitive abilities compared with the HC group, reflected in the MMSE, MoCA and AVLT scores.

Differences in ERC thickness, volume and surface area among the aMCI-s, aMCI-m, AD and HC groups were first assessed. As shown in Figure 1B, there was a significant difference in the ERC thickness across the four groups (*p* < 0.01). Post hoc analyses revealed a significant decreasing tendency in ERC thickness from HC to aMCI-s to aMCI-m to AD in both the left and the right hemispheres. Specifically, for the left hemisphere (F (3, 87) = 6.134, *p* < 0.001), with multiple comparison tests: *p* value of AD vs aMCI-s was 0.016, *p* value of AD vs HC was less than 0.001, *p* value of aMCI-m vs HC was 0.008, and *p* value of aMCI-s vs HC was 0.024. For the right hemisphere (F(3, 87) = 10.933, *p* < 0.001), with multiple comparison tests: *p* value of AD vs aMCI-m was 0.016, *p* value of AD vs aMCI-s was less than 0.001, *p* value of AD vs HC was less than 0.001, *p* value of aMCI-m vs aMCI-s was 0.027, and *p* value of aMCI-m vs HC was 0.005.

For the ERC volume, the AD group had significantly decreased volume on both sides compared with the HC group (right: *p* < 0.001; left: *p* = 0.002) and the aMCI-s group (right: *p* = 0.012; left: *p* = 0.016) after multiple comparisons (Figure 1C). In addition, the aMCI-m group also showed significantly decreased volume on both sides compared with the HC group (right: *p* = 0.002; left: *p* = 0.02; Figure 1C). However, as for the ERC surface area, no significant difference was identified among the four groups (*p* > 0.4; Figure 1D).

The relationship between ERC morphometric measurements and memory performance was assessed for HC, aMCI-s, aMCI-m and AD groups separately using partial correlation analyses (Appendix A). In the aMCI-m group, the AVLT scores correlated with ERC thickness (right: r = 0.723, *p* < 0.001) and volume (right: r = 0.796, *p* < 0.001; left: r = 0.645, *p* = 0.003). The MMSE scores correlated with ERC thickness (left: r = 0.600, *p* = 0.007). In addition, there was a correlation between MoCA scores and ERC thickness (left: r = 0.697, *p* = 0.001) and volume (left: r = 0.632, *p* = 0.004). For the HC, aMCI-s and AD groups, task scores did not correlate with any of the morphometric measurements.

The AUCs of single ERC indexes (including thickness, volume and surface area) are summarized in Appendix A. Relative to ERC surface area or ERC volume, ERC thickness is the best index to distinguish the four groups. The discriminating ability of ERC thickness and combination of ERC thickness and volume was compared (shown in Table 2). The AUC value of thickness and volume in combination was higher than that in thickness only.

Finally, the discriminating ability of combined ERC indexes (combination of ERC thickness and volume) was further compared to that of HP volume, which is the most studied and used MRI biomarker of AD. In terms of the classification accuracy of HC from aMCI-s and aMCI-m from AD, the AUC value of combined ERC indexes was higher than that of HP volume (Figure 2). In addition, the ability of combined ERC indexes was similar to that of the volume of HP in discriminating between HC and aMCI-m, HC and AD, aMCI-s and aMCI-m and aMCI-s and AD (Appendix A).

## 4. Discussion

In this study, alterations in ERC morphological indexes (including thickness, volume and surface area) were first assessed among AD patients and aMCI-m, aMCI-s and HC participants. The ERC thickness, rather than the ERC volume and surface area, showed a significant tendency in the conversion from aMCI to AD. Then, the AUC results demonstrated that combining ERC thickness and volume could better discriminate the four groups from each other than a single ERC index alone. Furthermore, relative to the volume of the HP, the combination of ERC thickness and volume had better discriminating capacity between HC and aMCI-s, as well as aMCI-m and AD.

ERC atrophy has been regarded as an early potential biomarker in patients with MCI and AD [10,26,27,28,29,30]. Our findings suggest that ERC thickness showed more significant changes than ERC surface area among AD, aMCI-m and aMCI-s, which was consistent with a previous study that showed AD appearing to have different effects on the thickness and surface area [13]. To be more precise, both hemispheres of ERC thickness showed a decreasing trend from HC to aMCI-s to aMCI-m to AD. These results add to the evidence that aMCI-m is more likely a transitional stage between aMCI-s and AD [5,6,17,31]. More importantly, our AUC results further prove that ERC thickness has superiority over ERC surface area and volume in discriminating among the four groups. Thus, on the basis of previous studies, our study found that aMCI-s and aMCI-m showed different magnitudes of decreased cortical thickness in the bilateral ERC relative to the HC group. Therefore, ERC thickness may serve as a potential diagnosis index in patients with aMCI.

ERC thinning is sensitive to the early pathological process of AD, which may be due to its own neurophysiological mechanism. On the one hand, early structural changes in AD are limited to specific laminae within the ERC (layer II is particularly vulnerable) [32]. On the other hand, the thickness of the cerebral cortex was calculated as the average distance between the gray/white boundary and the pial surface [33,34]. It likely represents cytoarchitectural features or many components of the neuropil, such as intra-cortical axons, dendrites, synaptic elements and glia [13,34,35,36]. It was reported that AD-related pathological alterations first resulted in synaptic neurodegeneration and then neuronal loss [37]. In addition, no significant neuronal loss in the ERC was detectable in cognitively normal participants, while a very severe neuronal loss was seen in the ERC of very mild AD cases [38]. On account of the reasons above, ERC thickness, rather than ERC surface area and volume, showed significant change even in the stage of aMCI-s.

Our results also reveal that ERC surface area was minimally affected in the conversion from aMCI to AD. To our surprise, the aMCI-s group even showed an increasing trend. This may be explained by compensation for ERC thinning [38]. The increase in ERC surface area autonomously compensates for ERC thinning in patients at the earliest preclinical stage (e.g., aMCI-s), whereas there is an absence of such compensation mechanism in patients at a late clinical stage (e.g., AD). However, the neurophysiological correlates of cortical surface area are less clear. ERC surface area may relate to local subcortical factors, such as subjacent white matter volume, or global factors, such as the head size [13]. Thus, regional analyses of cortical surface area must take into account the global effects of head size and brain size [13,34]. Previous work has shown that aging was related to reduced surface area, rather than AD [13,39]. In line with this, we found that ERC surface area was relatively unchanged in aMCI and AD after adjusting for head size.

By definition, ERC volume was a product of thickness and surface area [13,33,39]. The ERC volumetric decrease was the result of a combination of ERC thinning and ERC surface area change. There was no significant change in ERC surface area, even a slight increase in aMCI-s. That may explain why ERC volume atrophy was not significant in the subtypes of aMCI.

As discussed above, the alteration of different ERC morphological indexes varied among the four groups. Combining multiple ERC indexes (e.g., volume, thickness and surface area) may provide a complete understanding of progressive structural brain changes during the conversion of aMCI to AD. Moreover, the different morphological features had unique contributions to the classification of aMCI patients and healthy controls [34]. Thus, multi-parametric indexes may have the ability to detect subtle alterations in the progression of AD. The multivariate method, which combined certain indexes together, allows us to determine the relationships among different features beyond their individual values. Consistent with prior studies, the AUC results show that the combination of ERC thickness and volume further improved discrimination among the four groups.

Memory impairment is the earliest and most prominent symptom of aMCI and AD [40,41,42]. Since the medial temporal lobe structures (ERC and HP) are specialized for memory functions, alteration of the medial temporal lobe, especially volumetric loss of the HP, has been considered to be a key feature for early diagnosis of AD [43,44,45]. However, ERC atrophy may be more closely associated with the pathologic processes of AD than HP atrophy [38]. Thus, ERC atrophy could have an advantage over HP atrophy in discriminating among HC, subtypes of aMCI and AD.

The AUC results verify the above-mentioned assumption and show that the combination of ERC thickness and volume had a superior differential power than hippocampal volume for discriminating between HC and aMCI-s. The results also coincide with a pathologic study that stated that the pathology of AD starts in the ERC, providing in vivo evidence for the Braak stages (Stages 1 and 2 represent the entorhinal phase of the disease with minimal involvement of the hippocampus) [46]. In addition, the combination of ERC thickness and volume had a better discriminating capacity than the volume of the HP between aMCI-m and AD. According to a longitudinal MRI study, atrophy rates in AD were significantly higher for the ERC than for the HP [29]. Considering that the ERC was affected earlier and had a higher atrophy rate than the HP in AD [28,29], the combination of ERC multiple morphometric indexes should reflect more comprehensive information during the process from aMCI to AD. Therefore, it is quite understandable that the combination of ERC thickness and volume had an advantage regarding early diagnosis of aMCI and predicting conversion from aMCI to AD.

There were still some limitations in this study. First, although we used education level as a covariate for covariance analysis to reduce its impact on ERC evaluation, it is not as convincing as choosing subjects with a similar education level, but because our study was based on real clinical data, there was no perfect control obtained. Second, although the present study revealed a decreasing tendency in ERC thickness from HC to AD, this trend needs to be further confirmed by longitudinal studies. Finally, although ROC analyses can accurately reflect the authenticity of the diagnostic test, we may comprehensively select the appropriate diagnostic test according to many factors, such as the characteristics of subjects. To improve the classification accuracy and stability of results, we will use decision fusion techniques such as alpha integration as a continuation of this work [47].

## 5. Conclusions

In conclusion, this study demonstrated that ERC thickness, rather than ERC surface area or volume, showed a significant decreasing tendency from the aMCI group to the AD group. In the study, the combination of ERC thickness and volume had a superior power than any single ERC index alone or the volume of the HP for discriminating between HC and aMCI-s, as well as aMCI-m and AD. These findings suggest that ERC atrophy, particularly multi-index (combination of ERC thickness and volume), might be regarded as a promising candidate biomarker in the early diagnosis of aMCI as well as in the prediction of conversion from aMCI to AD.

## Figures and Tables

**Figure 1 brainsci-11-01129-f001:**
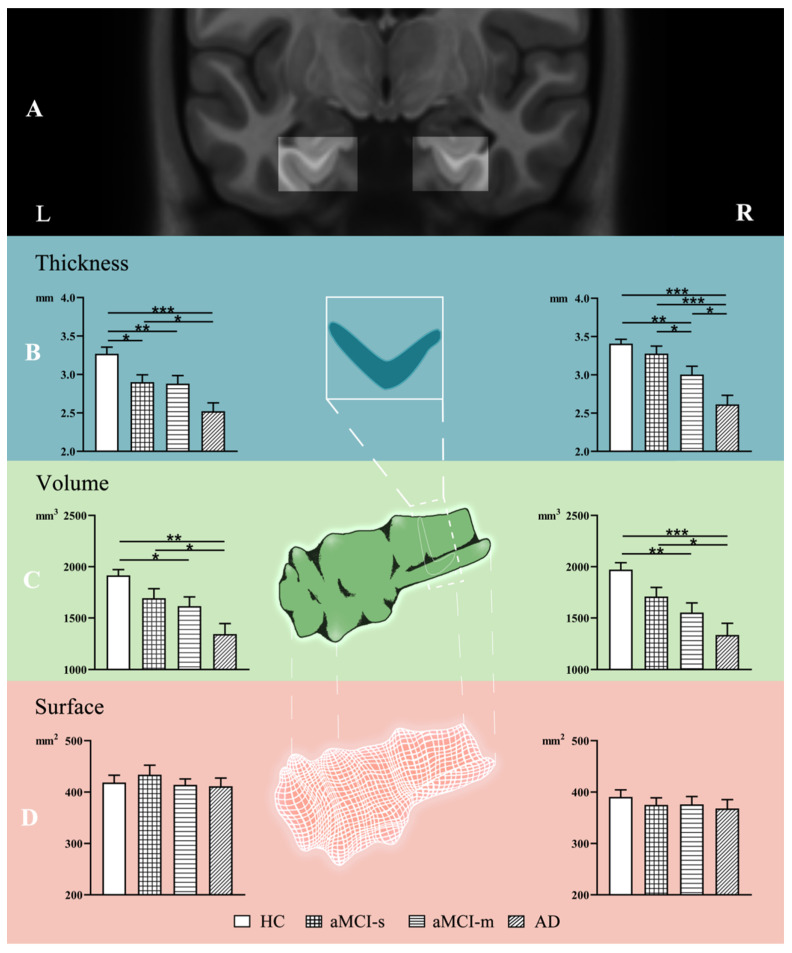
Alterations of ERC thickness, volume and surface area among HC, aMCI-s, aMCI-m and AD groups. The ERC location of left and right hemispheres in coronal slices (**A**). The figure shows group differences in ERC thickness (**B**), volume (**C**) and surface area (**D**). * *p* < 0.05; ** *p* < 0.01; *** *p* < 0.001. L: left, R: right.

**Figure 2 brainsci-11-01129-f002:**
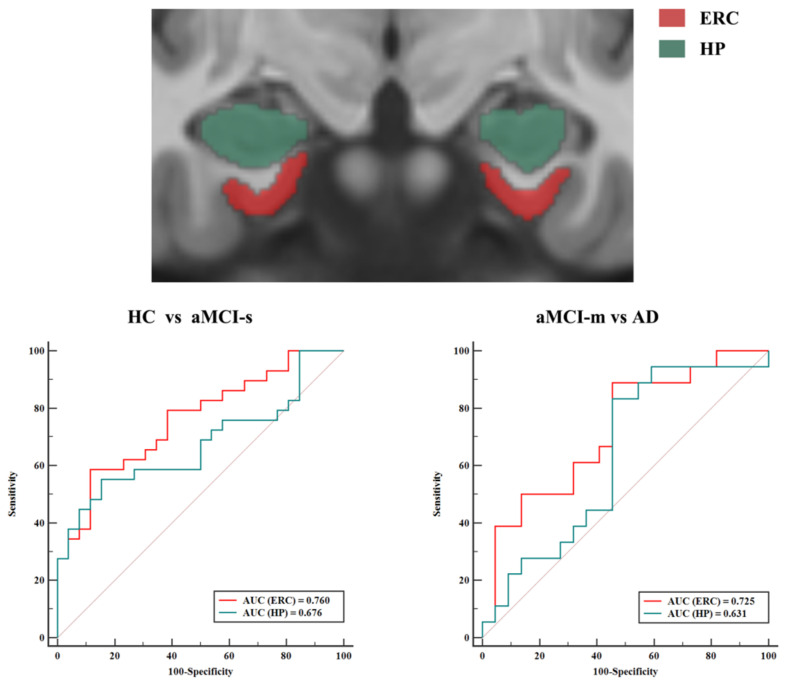
The ERC vs. HP in discriminating HC from aMCI-s and aMCI-m from AD. Red: entorhinal cortex; green: hippocampus.

**Table 1 brainsci-11-01129-t001:** Demographic and neuropsychological assessments of participants.

	AD Mean (SD)	aMCI-m Mean (SD)	aMCI-s Mean (SD)	HC Mean (SD)	*p* Value
Gender (Females/Males)	11/7	10/12	14/15	15/11	0.69 ^#^
Age	70.94 (9.77)	71.09 (8.41)	71.21 (6.48)	70.38 (5.36)	0.979 *
Education (years)	7.06 (3.69)	10.32 (3.72)	8.07 (3.85)	12.19 (3.26)	<0.001 *
MMSE	15.89 (7.05)	24.45 (4.04)	24.07 (3.47)	28.19 (1.47)	<0.001 *
MoCA	11.56 (5.35)	20.36 (4.47)	19.45 (4.24)	26.58 (1.70)	<0.001 *
AVLT	12.06 (7.21)	29.50 (11.12)	27.14 (5.74)	48.08 (9.29)	<0.001 *
CDR (0, 0.5, 1–2)	0.5 = 1, 1–2 = 17	0.5 = 22	0.5 = 29	0 = 26	<0.001 ^#^
BNT	11.94 (6.26)	23.36 (2.15)	27.76 (1.35)	28.96 (0.96)	<0.001 *
TMT	260.89 (49.53)	114.09 (29.88)	79.55 (23.35)	86.27 (34.51)	<0.001 *
CDT (0, 1, 2, 3)	2 = 6, 1 = 7, 0 = 5	1 = 6, 2 = 9, 3 = 7	2 = 5, 3 = 24	2 = 1, 3 = 25	<0.001 ^#^

Note: One-way ANOVA and chi-square analyses were applied to test for group differences. Statistical significance level was set at *p* < 0.05 (two-tailed). * The *p* value was obtained using one-way ANOVA. ^#^ The *p* value was obtained using chi-square test.

**Table 2 brainsci-11-01129-t002:** The AUCs of ERC thickness and ERC thickness and volume combined.

	Right Thickness	Left Thickness	Combined Thickness and Volume
HC vs. aMCI-s	0.558	0.711	**0.760**
HC vs. aMCI-m	0.736	0.729	**0.788**
HC vs. AD	0.908	0.876	**0.919**
aMCI-s vs. aMCI-m	0.647	0.505	**0.687**
aMCI-s vs. AD	0.824	0.711	**0.833**
aMCI-m vs. AD	0.694	0.707	**0.725**

The Bold emphasis the AUC value of thickness and volume in combination was higher than that in thickness only.

## Data Availability

The data that support the findings of this study are available on request from the corresponding author.

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
