# Peer review of "Magnetic Resonance Imaging Measurement of Entorhinal Cortex in the Diagnosis and Differential Diagnosis of Mild Cognitive Impairment and Alzheimer’s Disease"

_brainsci, 2021, doi:10.3390/brainsci11091129_

Round 1
Reviewer 1 Report
This is an interesting study to show that the entorhinal cortex is (more) affected in AD. The trend has pursued the hippocampus just because they thought fancy, but there was no reason why it has to be the case.
I wonder the authors thought about the relationship between the correlation of entorhinal volume/thickness and cognitive decline. There is a good chance that entorhinal reduction proceeds cognitive decline. just like other precursor symptoms known.. I don't believe it was discussed, but entorhinal volume may be a *biomaker* for early detection of AD. Sounds cool, right?
Author Response
Thank you very much for your helpful advice and suggestion. The relationships between ERC morphometric measurements and cognitive domain scores were examined using partial correlation analyses, with age, gender, and years of education as covariates with Bonferroni correction.
The relationship between ERC morphometric measurements and memory performance was assessed for HC, aMCI-s, aMCI-m and AD groups separately using partial correlation analyses. In aMCI-m group, the AVLT scores correlated with ERC thickness (right: r=0.723, P<0.001) and volume (right: r=0.796, P<0.001; left: r=0.645, P=0.003). The MMSE scores correlated with ERC thickness (left: r=0.600, P=0.007). In addition, there was a correlation between MoCA scores and ERC thickness (left: r=0.697, P=0.001) and volume (left: r=0.632, P=0.004). For the HC, aMCI-s and AD groups, task scores did not correlate with all morphometric measurements. (Please see line 145-147 and 195-202 for details of modification.)
Reviewer 2 Report
The objective of the paper is a study to classify magnetic resonance images (MRI) from Alzheimer’s disease patients for differential diagnosis of mild cognitive impairment. The paper demonstrate capabilities of different entorhinal cortex (ERC) features, and combination of those features, extracted from MRI data for classification. Results seem convincing. The data processing techniques employed are known (using MedCalc software), but clinical findings are very interesting from a practical standpoint. Thus, the contribution of the paper is fair. Literal presentation of the paper is good, but there is room for improvement in this aspect. Discussions are enlightening and rationale on conclusions is provided including limitations of covariance analysis. Some lines for future research work should be added. In summary, the following minor issues should be addressed in a revised version.
- Literal presentation has room for improvement. For instance, (i) last paragraph of page 3; line 194, there is a missing period. (ii) For readability, please add different markers to the curves of Figure 2. (iii) Page 9, line 308, change “Table 1” to “Table S1”. Thus, an English proofreading of the paper is recommended.
- The classification method implemented in the software package used for data processing should be briefly commented. Some limitations or bias in results could be derived from the classification method employed.
- Some lines for future research work both clinical and data analysis should be added. For instance, classification accuracy and stability of results might be improved using decision fusion techniques such as alpha integration. Such techniques would also facilitate a multimodal data fusion research that could be undertaken as a continuation of the work presented. I suggest the following reference: https://doi.org/10.1162/neco_a_01169.
Author Response
Comment 1:The classification method implemented in the software package used for data processing should be briefly commented. Some limitations or bias in results could be derived from the classification method employed.
Response: Thank you very much for your favorable advice and suggestion. We have added comment in the statistical analyses as follows: “the receiver operating characteristic (ROC) analyses were performed that shows the diagnostic ability of a binary classifier as a function of its discrimination threshold.” (Please see page 4, paragraph 1 for details of modification.)
Comment 2: Some lines for future research work both clinical and data analysis should be added. For instance, classification accuracy and stability of results might be improved using decision fusion techniques such as alpha integration. Such techniques would also facilitate a multimodal data fusion research that could be undertaken as a continuation of the work presented.
Response: It will be of great help to our continued work after reading your recommended article. We have added both clinical and data analysis in the discussion as follows: “Finally, although ROC analyses can accurately reflect the authenticity of the diagnostic test, we may comprehensively select the appropriate diagnostic test according to many factors, such as the characteristics of subjects. To improve the classification accuracy and stability of results, we will use decision fusion techniques such as alpha integration as a continuation of this work.” (Please see page 9, paragraph 4 for details of modification.)
Comment 3:Literal presentation has room for improvement. For instance, (i) last paragraph of page 3; line 194, there is a missing period. (ii) For readability, please add different markers to the curves of Figure 2. (iii) Page 9, line 308, change “Table 1” to “Table S1”. Thus, an English proofreading of the paper is recommended.
Response: Thank you very much for your reminder. We asked the help from a native English speaker to improve literal presentation.
We have add different markers to the curves of Figure 2. We also change “Table 1” to “Table S1”.